# Towards Meta-Models for Automated Interpretability

## Abstract

Previous work has demonstrated that in some settings, the mechanisms implemented by small neural networks can be reverse-engineered. However, these efforts rely on a manual approach that cannot easily be applied to networks with billions of parameters. To investigate a potential avenue towards scalable interpretability, we show it is possible to use *meta-models*, neural networks that take another network's parameters as input, to learn a mapping from transformer weights to human-readable code. We build on Tracr (Lindner et al. 2023) to synthetically generate transformer weights that implement known programs in the RASP language (Weiss et al. 2021), then train a transformer to extract RASP programs from weights. Our trained compiler effectively extracts algorithms from model weights, reconstructing a fully correct algorithm 60% of the time.

## 1 Introduction

Neural networks are typically black boxes; we know that they are able to perform a task (image recognition, language modeling, etc.), but we do not know *how* they perform it. In this work, we approach the problem of extracting a full description of the computations implemented by a neural network and displaying it in a human-readable form. We propose to train a neural network (the **meta-model**) to produce a full description of the algorithm implemented in a small transformer encoder (the **base model**) when given the base model's parameters as input.

A challenge for methods aiming to extract an algorithm description from a base model is that we typically do not have access to the ground truth algorithm. Thus it is difficult to evaluate or train a method for extraction. To overcome this challenge, we introduce a dataset of 1.6 million base models that implement known programs. We leverage RASP (Weiss et al. 2018), a programming language designed as a computational model for transformers, and Tracr (Lindner et al. 2023), a compiler that compiles RASP programs to transformer weights.

**Contributions:**

- We design **rasp-gen**, a sampler that generates valid RASP programs, and use it to construct a dataset consisting of 1.6 million RASP programs and corresponding model weights. (Section 3)
- We train a transformer meta-model to recover RASP programs directly from model weights. (Section 3)

The trained meta-model accurately recovers RASP programs 60% of the time on an i.i.d. test set. The meta-model is also able to recover a hand-written sorting algorithm, not generated by the program sampler and confirmed to not be present in the training set (Figure 2).

## 2 Background: RASP and Tracr

The *Restricted Access Sequence Processing* language (RASP) is a domain-specific programming language developed by Weiss et al. (2021) to provide a computational model for an encoder-only transformer. A RASP program receives two inputs: a length-$n$ sequence of tokens and corresponding positional indices ranging from 0 to $n - 1$. The inputs are then transformed by RASP operations that

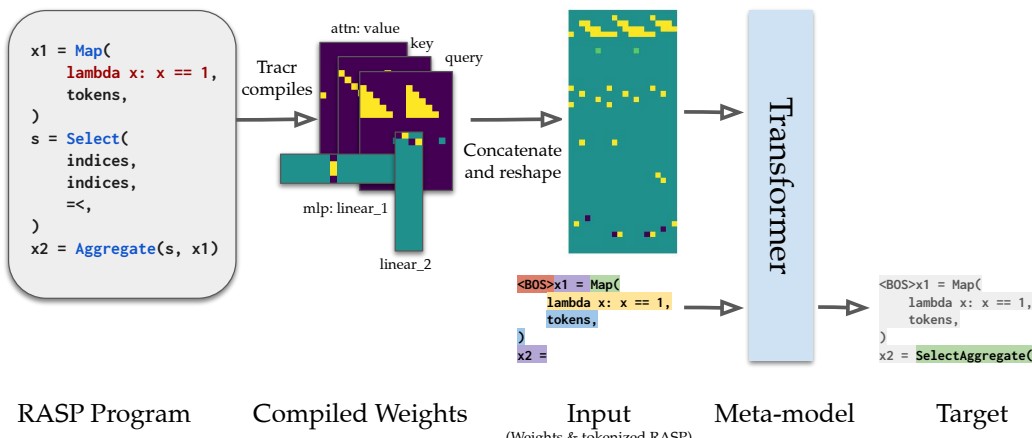

RASP Program     Compiled Weights     Input     Meta-model     Target
(Weights & tokenized RASP)

Figure 1: We train a meta-model (a transformer decoder) to take base model parameters as input and output the RASP program implemented by the base model. We concatenate and reshape the base model parameters for model input. Separately, we tokenize the RASP program. Weights and token embeddings are concatenated to form an input sequence. During training, the meta-model learns to predict the next token in a RASP sequence; at test time, it generates RASP programs autoregressively.

correspond to either MLP or attention layers in a transformer. RASP syntax differs slightly between implementations; we build on the RASP implementation by Lindner et al. (2023), in which RASP consists of five basic operations. These operations can be distinguished by their correspondence to either MLP or attention layers:

**Elementwise mappings (MLP layers).** RASP programs can implement arbitrary elementwise mappings on sequences. The Tracr implementation of RASP uses the `Map` and `SequenceMap` operation to implement such mappings; for example

$$\texttt{SequenceMap(f, a, b)}$$

for $\texttt{f}(x, y) := x + y$ returns the elementwise sum of sequences `a` and `b`. The `Map` operation works just like `SequenceMap`, but instead operates on a single sequence, e.g. to implement an elementwise $f(x) = x^2$.

**Select-Aggregate operations (attention layers).** To move information between sequence elements, RASP uses `Select` and `Aggregate` operations: given a boolean predicate `predicate` and two sequences `a` and `b`, the `Select` operation returns a boolean 'selector' matrix:

$$\texttt{Select(a, b, pred)} := (\texttt{predicate}(\texttt{a}_i, \texttt{b}_j))_{ij<n}.$$

To reduce a selector matrix to a sequence, the `Aggregate` operation takes as input a selector and a sequence, and returns the average of the input sequence weighted by the nonzero elements in the selector.

A simple example program in RASP is available in Figure 2. For more detail on RASP, please refer to Weiss et al. (2021) and Lindner et al. (2023). Additionally, Figure 9 may be helpful for understanding the `Aggregate` operation.

**Tracr.** Tracr (Lindner et al. 2023) is a compiler for a large subset of RASP. Given a RASP program, Tracr outputs a set of transformer weights that implement the RASP program. To compile a RASP program, Tracr first computes the value set (i.e. the set of possible values) of sequences in intermediate layers using a Python implementation of RASP, then converts each RASP operation into either an MLP layer or an attention head that implements the same mapping. Where possible, layers are then merged and stacked, forming a sequence of MLP and multi-head attention layers.

Tracr distinguished two kinds of sequences: categorical and numerical. Categorical sequences are assumed to be discrete-valued. During compilation, Tracr transforms RASP operations on categorical

```
input: tokens, indices
sel = Select(tokens, tokens, <)
sop0 = SelectorWidth(sel)
sel = Select(sop0, indices, ==)
return Aggregate(sel, tokens)
```

Figure 2: Example RASP program recovered by our decompiler at test time. This program from Lindner et al. (2023) uses attention operations to sort the input. When compiled by Tracr, the operations Select and SelectorWidth are implemented in the first attention layer, and the operations Select and Aggregate are implemented in the second attention layer. The training set is deduplicated of any instances of this particular program, so it is an unseen example.

sequences into a lookup table. Operations on float-valued (numerical) sequences are instead compiled into a a piecewise-linear MLP mapping obtained via solving an optimization problem; thus numerical operations are inexact when compiled. Tracr provides a special primitive `LinearSequenceMap` which functions like `SequenceMap`, but is constrained to weighted sums of the input elements which can be compiled efficiently without the need for fitting an approximation.

Tracr places some limitations on the use of numerical sequence variables. While the `Aggregate` operation accepts numerical sequence inputs, `Select` operation only accepts categorical inputs; that is, values may be numerical while keys and queries must always be categorical. In addition, numerical inputs to `Aggregate` must take values in $\{0, 1\}$, thus constraining float-valued attention outputs to the interval $[0, 1]$.

A major motivation for the development of Tracr is its potential as tool for interpretability; for example, Lindner et al. (2023) use compressed Tracr-compiled models to study a neural network's tendency to compress a large number of sparse features using superposition (Elhage et al. 2022).

## 3 EXPERIMENTS

We train a meta-model to map transformer parameters obtained via Tracr to the corresponding RASP programs. Code and datasets will be made available under an open-source license.

### 3.1 TRAINING A DECOMPILER FOR TRACR

We generate a dataset of 1.6 million RASP programs compiled using Tracr and train a meta-model to map transformer weights directly to RASP code, effectively training a decompiler for Tracr. This experiment functions as a proof of concept to show that meta-models are able to reverse-engineer algorithms implemented in compiled transformers.

**Sampling RASP Programs** In order to generate our dataset, we need to sample random RASP programs. To sample a program, we sequentially sample an operation from the set {Map, SequenceMap, LinearSequenceMap, Select, Aggregate, SelectorWidth} while keeping track of available sequence variables, starting from the two input sequences (input tokens and positional indices). To make sure that sampled programs are nontrivial we filter out programs that are constant or equal to the identity on a set of test inputs. We further filter out programs in the subset of RASP not supported by the Tracr compiler. After we finish sampling and filtering programs, the resulting programs are tokenized and deduplicated.

**Tokenizing.** In order to cast decompilation as a sequence prediction task, we tokenize the RASP language via a vocabulary of 105 tokens consisting of variable names, operations, encodings, a set of 60 possible elementwise mappings, and a set of possible predicate functions. Since every `Select` operation is always followed either by an `Aggregate` or a `SelectorWidth`, we fuse `Select`s with the subsequent operation when tokenizing; that is,

```
SelectAggregate(x, y, pred, z) = Aggregate(Select(x, y, pred), z)
```

For example, the RASP program in Figure 2 is tokenized as follows (line breaks added between layers for clarity):

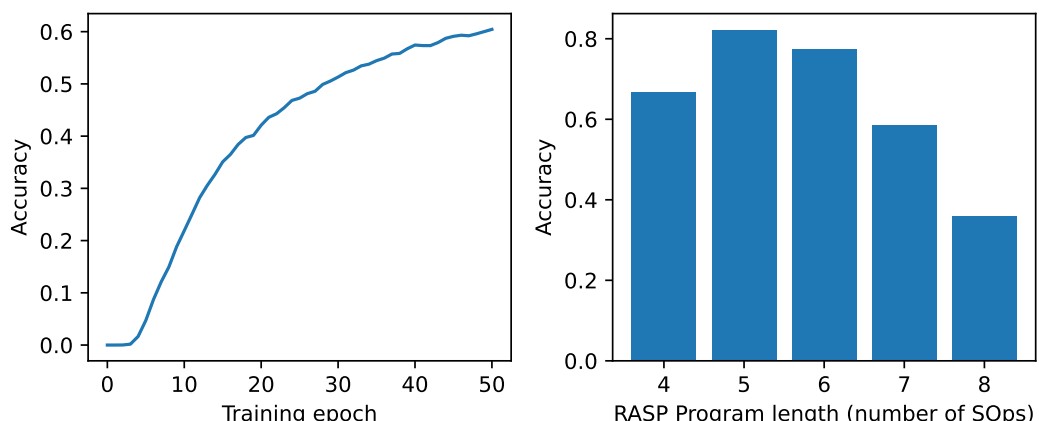

Figure 3: Accuracy (fraction of RASP programs recovered perfectly by the learned decompiler). **Left:** Validation accuracy across training time. **Right:** Final test accuracy by program length. Length is measured by number of sequence operations (e.g. `Map`, `Aggregate`, etc.) in a RASP program. To count as 'recovered', the decompiler meta-model needs to correctly predict the entire program. When tokenized, most RASP programs are 30-60 tokens long and when compiled result in a model of 5-10 layers and between 1,000-60,000 parameters.

1. `BOS sop_0 categorical SelectorWidth tokens tokens LT EOO EOL`

2. `EOL`

3. `sop_1 categorical SelectAggregate sop_0 indices EQ tokens EOO EOL`

4. `EOL EOS`

Note that `BOS` and `EOS` tokens mark the beginning and end of the entire program, while `EOO` marks the end of an operation and `EOL` marks the end of a layer. Since this particular program does not include any `Map` or `SequenceMap` operations, the MLP layers are empty.

**Base model dataset.** We generate a dataset of 1.6 million RASP programs via the procedure described above and use Tracr to compile every program to a set of transformer weights. This results in a dataset consisting of tuples $(P, W)$, where $P$ is a RASP program and $W$ is the corresponding set of transformer weights. We deduplicate this dataset after generation. Generated programs contain between 4 and 9 SOps (sequence operations), and the compiled transformers are between 3 and 10 layers deep. Every compiled transformer contains between 600 and 65,563 weights.

Compilation using Tracr is computationally cheap; compiling a single model takes under five seconds on average on a single CPU, and to generate the full dataset we used approximately 1000 CPU-hours (CPU cores $\times$ hours worked). This stands in contrast to the cost of training thousands of base models as typical for previous work on meta-models (Eilertsen et al. 2020; Schürholt, Kostadinov, et al. 2021). For instance, it cost us 1200 A100-hours to generate the dataset in Appendix A.

**Meta-model training.** We cast decompilation as a supervised next-token prediction task. For input to the meta-model, we flatten the base model weights and pad them to a fixed length $m$, then reshape them into an array $w \in \mathbb{R}^{m/d \times d}$ where $d$ is the embedding dimension of the meta-model. The tokenized RASP operations are padded to a fixed length $r$ and embedded via a linear layer as is standard in language modeling. We then concatenate the weights and the RASP program, resulting in an input array $x \in \mathbb{R}^{(m/d+r) \times d}$. In our experiments we pick $m = 65,536$, $d = 256$, and $r = 128$.

We train the meta-model to predict the next token in the RASP program via a standard cross-entropy loss. At test time, we generate an entire RASP program autoregressively: we condition the trained model on a set of base model parameters and perform consecutive model calls to generate a RASP program.

**Uniqueness of compilation and decompilation.** In general, different sets of model weights can implement the same function, e.g. due to symmetries in MLP layers. Similarly, the Tracr compiler may return different sets of weights given the same RASP program, since numeric MLP operations are compiled via a piecewise linear approximation found by solving a nondeterministic optimization problem. In the other direction, two distinct RASP programs may compute the same function. If this is the case, the Tracr compiler does not guarantee that two RASP programs compile to distinct models. Thus it is possible that in some situations, our decompiler must choose between two RASP programs that both validly describe the base model weights. However, we have not been able to find such cases in our dataset.

**Results.** Our results are reported in Figure 3, and we display an example of a short reconstructed program in Figure 2. We evaluate the meta-model on a i.i.d. test set of programs which we split off after deduplication, so it is guaranteed to consist of unseen examples. On this test dataset the decompiler is able to decompile $60\%$ of programs without errors. On a per-token level it achieves an accuracy of $98.3\%$; a tokenized RASP program typically consist of between 30 and 60 tokens. Unsurprisingly, the accuracy degrades significantly with program length, dropping from $80\%$ on programs consisting of 5 operations down to $26.3\%$ for programs consisting of 8 operations. We also evaluate on a handcrafted RASP program that sorts an input sequence (Figure 2), which we ensure is unseen during training.

## 3.2 DECOMPILING FROM NON-SPARSE WEIGHTS

Weights obtained by compiling a RASP program via Tracr are dissimilar from weights obtained via training by gradient descent. Not only are Tracr-compiled models highly sparse, they also represent sequence variables (i.e. internal activations) in a disentangled fashion, as every RASP variable is represented by a separate linear subspace in the residual stream.

In fact, Friedman et al. (2023) show that a significant subset of Tracr-compiled models (those consisting only of categorical sequence variables) can be mapped to RASP code via a hand-crafted algorithm. While our dataset is more challenging, as it includes compiled models that operate on numerical variables, it is clear that decompiling Tracr is an easier problem than extracting algorithms from trained transformers in general. To account for this sparsity problem, we run a second experiment to show that our meta-model is still able to recover accurate RASP programs from non-sparse models.

The key to our approach is the observation that it is possible to apply a linear transformation to transformer weights without modifying the output. If $\boldsymbol{A} \in \mathbb{R}^{d \times d}$ where $d$ is the dimension of the residual stream, then consider modifying a model such that it applies $\boldsymbol{A}$ to activations before writing them to the residual stream and $\boldsymbol{A}^{-1}$ before reading from it, leaving the final output unchanged. Since every layer reads from and writes to the residual stream linearly, it is enough to multiply every weight matrix by $\boldsymbol{A}$ or $\boldsymbol{A}^{-1}$ as appropriate, resulting in a new set of transformer weights. Given a transformer with sparse weights, we can therefore construct a set of dense weights with the same outputs by sampling a random orthogonal matrix and applying it to the sparse weights.

Finally, we use PCA to learn a linear projection $\boldsymbol{B} \in \mathbb{R}^{d \times d'}$ to compress the original activations of size $d$ to a smaller dimension $d' < d$. We apply $\boldsymbol{B}$ in the same way as the sampled orthogonal matrix, multiplying weights by $\boldsymbol{B}$ or $\boldsymbol{B}^T$ as appropriate. This does not necessarily leave the output fully unchanged, but if $d'$ is not too small the outputs of the compressed model are equal on $> 99\%$ of inputs.

The purpose of this compression is to ensure that activations are not disentangled. In Tracr, RASP sequence variables are represented as orthogonal directions in the residual stream of a compiled model. In contrast, a common observation in trained transformers is that a model learns to make use of more features than can be orthogonally represented. Thus compressing the residual stream helps make our testbed more realistic.

**Base model dataset.** We construct a dataset of $780,000$ program-model pairs $(P, W)$ as in Section 3.1, keeping program length fixed at 5 RASP operations. For every datapoint, we then apply the de-sparsification procedure described above: we first multiply weights by a random orthogonal matrix, and then create a set of compressed weights $W'$ by applying a compression matrix obtained via PCA.

```
input: tokens, indices
select_1 = Select(tokens, tokens, predicate=GEQ)
select_2 = Select(tokens, tokens, predicate=GT)
select_3 = Select(tokens, indices, predicate=NEQ)
selector_width_1 = SelectorWidth(select_1)
selector_width_2 = SelectorWidth(select_2)
selector_width_3 = SelectorWidth(select_3)
sequence_map_1 = SequenceMap(lambda x, y:  x * (y + x) % 5,
selector_width_1, selector_width_2)
map_1 = Map(lambda x:  x < 0, selector_width_3)
select_4 = Select(sequence_map_1, selector_width_1, predicate=GEQ)
aggregate_1 = Aggregate(select_4, map_1)
return aggregate_1
```

Figure 4: A random RASP program sampled by our generator

**Results.** We train a new meta-model in the same way as in Section 3.1. However, instead of training on base models with sparse weights as returned by Tracr, we train on base models with weights compressed to be dense as described above. We then evaluate it on an i.i.d. test set of size $25,000$. On this test set the meta-model is able to decompile $77\%$ of programs without errors. On a per-token level it achieves an accuracy of $99\%$. Note that as the postprocessing to avoid sparsity is expensive, unlike in Section 3.1 we keep program length fixed at 5 operations.

## 4 LIMITATIONS

**Models obtained by Tracr are dissimilar from trained models** The models we train on tend to be compiled from simple RASP programs with no more than a few (1-5) RASP operations per layer and less than 10 total. It is likely that most transformers trained in realistic settings do not have a short representation in RASP.

**Training dataset.** We have chosen a task such that it is easy to generate a training dataset for the meta-model, and for which a loss function is easily evaluated. For Tracr in particular it is computationally cheap to generate hundreds of thousands of programs, and a ground truth explanation is readily available via the RASP program. It is likely to be harder to generate training data for real-world interpretability tasks. In addition, our meta-models tend to be larger than the base models they are trained on by about a factor of 10-100, which would be prohibitive for very large base models.

**We use a black box to interpret a black box.** Interpretability research can broadly be classified into two kinds of approaches: those that *generate* explanations, and those that *verify* explanations. We show that meta-models can be used to generate explanations, but do not address the problem of verifying the explanations produced by the meta-model. Without any means of verification, this approach cannot provide guaranteed assurances about the base models analyzed.

## 5 RELATED WORK

**Meta-models.** While to our knowledge we are the first to use the term *meta-model* in a paper, the idea of using neural networks to operate on neural network parameters is not new. A line of work focuses on training an autoencoder meta-model on a dataset of neural network weights (Schürholt, Kostadinov, et al. 2021; Schürholt, Knyazev, et al. 2022). The trained encoder can be used as a feature extractor to predict model characteristics (such as hyperparameters), and the decoder can be used to sample new weights, functioning as an improved initialization scheme. In earlier work, Eilertsen et al. (2020) train a meta-model to predict base model hyperparameters such as learning rate and batch size. Our meta-model architecture is simpler and outperforms both works on all tasks we tested

(Appendix A). Although we improve on the state-of-the-art, we don't include these results in the main text because our focus is on automating interpretability rather than hyperparameter prediction.

**Extraction.** Weiss et al. (2018) algorithmically extract a representation of an RNN as a finite state automaton. This is similar to our work because we are also interested in extracting a full description of the computation performed by a transformer (Section 3); the main difference is that we learn an extraction method (rather than using a fixed algorithm), and that we work with compiled rather than trained models. Other works that have extracted programmatic representations of functions implemented by trained neural networks include Cai et al. (2017) and Mikulik et al. (2020). More recently, Friedman et al. (2023) show it is possible to deterministically extract a RASP-like description from transformer parameters trained to operate on categorical variables in a fashion explicitly inspired by Tracr.

**Hypernetworks.** Hypernetworks (Ha et al. 2017) are neural networks that generates the weights of another network (usually called the 'main' network). Typically, a hypernetwork takes a layer index and other layer information as input and outputs the weights for that layer, thus achieving a kind of relaxed weight sharing between layers. One trains a hypernetwork by jointly back-propagating through the main network and the hypernetwork. Hypernetworks are related to meta-models in that they operate on weights directly. They are different in that hypernetworks return weights as output, whereas meta-models take weights as input.

**Interpretability.** The field of interpretability studies the workings of machine learning models, with the goal of making the outputs and behaviour of these models more understandable to humans (Doshi-Velez and Kim 2017; Lipton 2018). While there is no universally agreed-upon definition of interpretability, in the context of this work, we focus on the particular problem of *mechanistic interpretability*, which aims to fully reverse engineer the learned algorithm implemented by a neural network. Despite the supposed black-box nature of neural networks, the field has had some noteworthy successes understanding network internals (Cunningham et al. 2023; Bricken et al. 2023), in one setting fully understanding the exact algorithm implemented by a network (Nanda et al. 2023). However, so far these successes have mostly been restricted to relatively small models, and either only consider models trained on toy tasks or limited aspects of a model's behavior. Other recent work on mechanistic interpretability includes tracking chess knowledge in AlphaZero (McGrath et al. 2022), locating a circuit responsible for a specific grammatical task in GPT-2 (Wang et al. 2022), and the study of superposition in transformers (Elhage et al. 2022).

There have been a number of proposed approaches to *automating* mechanistic interpretability, including automated circuit ablation (Conmy et al. 2023) and verification of circuit behavior (Chan et al. 2022). Both of these works study automatic verification of hypotheses, but don't propose a method for automatic *generation* of hypotheses. A different approach to automated interpretability is to use LLMs to annotate neurons based on dataset examples (Bills et al. 2023; Foote et al. 2023). While this allows for the automatic generation of hypotheses, these hypotheses are written in natural language and thus hard to verify and likely unreliable.

## 6 FUTURE WORK

Our work provides a first proof-of-concept for the approach we propose: using meta-models to automate aspects of mechanistic interpretability. A number of challenges remain before this approach can be applied practically.

**Scaling Meta-Models.** A challenge in training meta-models is that training data is either synthetic and thus potentially unreliable (such as Tracr-compiled models), or expensive to generate (such as when generating a large dataset of trained base models). This is especially problematic for large state-of-the-art models (e.g. LLMs), since training hundreds or thousands 'frontier' models is not feasible. There are a number of potential avenues to effectively scaling up meta-models to process large input models. Questions include: (1) Can large-scale pre-training on a base model zoo (e.g. doing masked weight prediction, or contrastive learning) improve performance? (2) Can a meta-model trained on smaller base models generalize to larger base models, implying that neural circuitry is consistent

across scale? (3) Can meta-models be readily applied to problems that only require processing a small part of a base-model at a time?

**Transformer Reverse-Engineering.** Tracr-compiled models are relatively sparse compared to trained transformers. We suggest a couple steps to approach general transformer reverse-engineering. (1) Can meta-models reverse-engineer transformers obtained from a more realistic variant of the Tracr compiler featuring a compressed residual stream and SGD-trained weights? [1] (2) Can a meta-model trained on Tracr-compiled models generalize to transformers trained on the inputs and outputs of similar RASP programs? If transfer from Tracr-compiled models is helpful, it may be possible to cheaply generate large training sets for meta-models.

**Creating Hypotheses for Causal Scrubbing.** Causal scrubbing (Chan et al. 2022) is a technique for evaluating whether a simplified, human-legible computational graph is an accurate model of a given neural network circuit. Can a simple dataset be constructed with one-to-one pairs of (network circuit, equivalent computation graph)? Can a meta-model be trained on this dataset and learn to map circuits to mechanistic explanations?

**Classifying Attention Heads in LLMs.** Recent work in mechanistic interpretability has associated specific functions to attention heads in LLMs.[2] Can a meta-model be trained to identify the functions of attention heads using relatively few labeled examples? Operating on one head at a time has numerous benefits, as the meta-model need only process a small part of the input model, and a single large input model can produce many labeled training examples.

**Automated Verification of Interpretations.** Can a meta-model be trained to output not only a programmatic description of the base model, but also evidence or proof that this description is accurate? One approach would be to train a meta-model to adversarially suggest examples which might *disprove* any proposed equivalence between a model and an interpretation.

## 7 CONCLUSION

Scaling is currently a major bottleneck for mechanistic interpretability. The current state-of-the-art requires substantial human labor by researchers to understand a model, and may remain infeasible for many large models in deployment. We propose to use transformers, which show favorable performance scaling, as "meta-models"—models that take other models weights as input—that can be trained to perform interpretability tasks. The method is general: we apply it to generating human-readable code from neural networks, but it is very flexible; for example in Appendix A we apply our meta-model to the task of predicting hyperparameters from weights. Despite its generality, it performs well: beating prior work on both hyperparameter prediction and successfully recovering the majority of RASP instructions from Tracr-compiled transformer weights.

Our work indicates the potentially broad applicability of meta-models in the circumstances where it is possible to construct an appropriate supervised training set of models and interpretations. Future work may extend meta-models to more complex and more useful tasks.

---

[1] See Section 5 and the Appendices A.2 and F of Lindner et al. (2023).

[2] For instance, Name Mover and Backup Name Mover heads for the Indirect Object Identification task found by Wang et al. (2022).

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

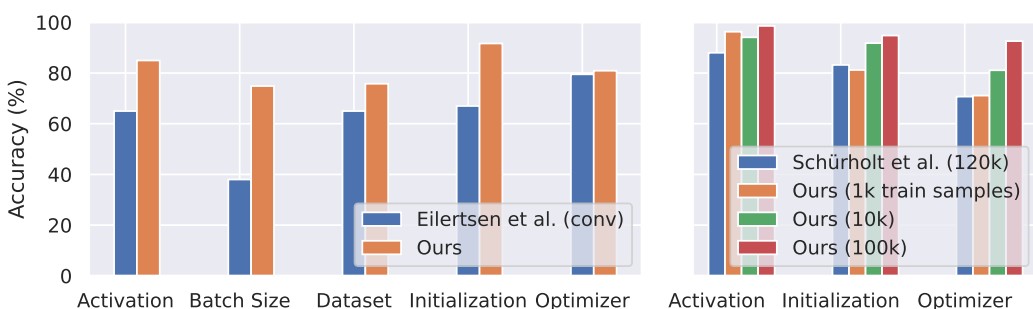

Figure 5: Comparison with prior meta-model work (Eilertsen et al. 2020; Schürholt, Kostadinov, et al. 2021). The task is to classify map neural network weights based on hyperparameter values. Despite not adapting our method to the task at all, we outperform prior work. This is true even when we train on far less data—for example, we match or outperform Schürholt, Kostadinov, et al. (2021) (right) using 100 times fewer training samples.

## A   COMPARISON WITH PRIOR META-MODEL WORK

Past work has applied meta-models (that is, neural networks that take weights of other neural networks as input) to a variety of tasks. To sanity check our choice of meta-model architecture as well as our methods for preprocessing network parameters for model input, we compare against Eilertsen et al. (2020) (henceforth EJR) and Schürholt, Kostadinov, et al. (2021) (henceforth SKB), who both train a meta-model to predict hyperparameters used to train base models.

The main difference between our meta-model and EJR/SKB is that we use a simple transformer encoder as meta-model. In order to compare against EJR/SKB, we adapt our meta-model to the classification setting by removing all causal attention masks and attaching a single linear layer to the output at position 0, from which we decode the logits for classification.

EJR use a CNN meta-model to predict (from the base model weights) the dataset, batch size, augmentation method, optimizer, activation function, and initialization scheme used to train the base model. They use two datasets: one where the architecture (and thus the size) of the base models are fixed, and another where the base models have variable size. We recreate their second dataset as it is the more general setting. We follow their dataset generation procedure, training CNNs with random selections of the hyperparameters listed above.

The setting of SKB is similar but differs in a few respects. SKB use a fixed model size for the base models, a smaller set of hyperparameters for classification, and an autoencoder architecture as the meta-model. The autoencoder is first pre-trained in an unsupervised manner to reconstruct neural network weights. After pretraining, the encoder plus an extra linear layer is fine-tuned to perform the classification task. While pretraining on large datasets is a promising direction, we chose to train a classifier directly.

**Base model dataset.**   We create two base model datasets corresponding to the experimental setups in EJR and SKB respectively. For comparison to EJR, we train 10,000 CNNs while randomizing the model size, dataset, batch size, augmentation method, optimizer, activation function, and initialization scheme used to train the base model. For comparison to SKB, we replicate their dataset construction and train 30,000 CNNs on each of MNIST, FashionMNIST, CIFAR-10, and SVHN while randomizing the optimizer, activation function, and initialization scheme used to train the base model.

We match the dataset size and composition for both EJR and SKB, including randomizing hyperparameters in the same way. The only difference in our setup is in the comparison to EJR, where we use fewer augmentations when training the CNN base models. This is because EJR use an large set of augmentations that is hard to replicate. We discuss this difference more in the appendix. We also provide more details on both base model datasets in the appendix. Training the CNNs for

these datasets used approximately 1200 A100-hours, while meta-model training used around 15 A100-hours. We release both base model datasets.[3]

**Meta-model training.**    For every hyperparameter (activation, batch size, and so on), we train a meta-model (a decoder-only transformer) to classify base model (input) weights based on the values of the hyperparameter. For example, the meta-model might predict the kind of activation function used (ReLU, ELU, Sigmoid, or Tanh). All meta-models are trained the hyperparameter classification task in a supervised fashion.

To prepare the CNN weights for model input we flatten them into a single vector of length $800,000$ by truncating or padding depending on the size of the base model. We then reshape the weights to form a sequence $x \in \mathbb{R}^{256 \times 3125}$ and embed $x$ via a linear layer. To use the transformer outputs for classification, we attach a single linear layer to the output at position 0.[4]

As EJR use a 1-dimensional CNN as meta-model, they are restricted to training on a 5,000-long randomly chosen segment of the flattened weights. As a transformer meta-model scales more easily, we only truncate base model weights past 800,000 parameters.

The results are visible in Figure 5. We outperform EJR and SKB in every category, sometimes substantially. While these problems are not clearly valuable from an interpretability standpoint, they show that our proposed meta-model architecture readily solves extant tasks and beats the state-of-the-art.

## A.1    Details

### A.1.1    Comparison with Eilertsen et al. (2020)

Eilertsen et al. (2020) use a CNN meta-model to predict (from the base model weights) the dataset, batch size, augmentation method, optimizer, activation function, and initialization scheme. They have two settings: one where the architecture (and thus the size) of the base models are fixed, and another where they are allowed to have variable size. We focus on the second, more general setting. We replicated their dataset generation procedure, training a dataset of 40,000 CNNs via a random search across hyperparameters and datasets (MNIST, CIFAR-10, SVHN, STL-10, Fashion-MNIST).

The base models were trained with the following hyperparameters. For meta-model training, every hyperparameter corresponds to a classification task. For example, dataset prediction is a 4-way classification task.

- Dataset: MNIST, CIFAR-10, CVHN, Fashion-MNIST,
- Batch size: 32, 64, 128, 256
- Optimizer: Adam, RMSProp, MomentumSGD
- Activation: ReLU, ELU, Sigmoid, Tanh
- Initialization: Constant, RandomNormal, GlorotUniform, GlorotNormal

### A.1.2    Comparison with Schürholt, Kostadinov, et al. (2021)

The setting of Schürholt, Kostadinov, et al. (2021) is similar to Eilertsen et al. (2020). The base model dataset consists of classifiers trained on four datasets: MNIST, FashionMNIST, CIFAR-10, and SVHN. Schürholt, Kostadinov, et al. (2021) train 30,000 base models on each of the four datasets, then train a meta-model classifier to detect hyperparameters (activation function, initialization scheme, and optimizer) from the base model weights. While Schürholt, Kostadinov, et al. (2021) train a separate meta-model on each dataset, we simply train one model and compare against the average performance over the four datasets.

The base models were trained with the following hyperparameters. For meta-model training, every hyperparameter corresponds to a classification task. For example, dataset prediction is a 4-way classification task.

---

[3]URL redacted for anonymity.

[4]This extra classification head is a standard trick and is used e.g. in Vision Transformers (Dosovitskiy et al. 2020).

- Activation: ReLU, Tanh

- Initialization: XavierNormal, HeNormal, Orthogonal, RandomNormal, TruncatedNormal

- Optimizer: Adam, RMSProp, SGD

# B  META-MODEL TRAINING

## B.1  TRANSFORMER TRAINING

We use the following hyperparameters for meta-model training in Section 3.

- Hidden dimension: 256

- Number of attention heads: 4

- Number of layers: 6

- Query size: 256

- MLP hidden size: 1024

- Dropout rate: 0 (no dropout)

- Learning rate: 0.0005

- Weight decay: 0.0001

- Batch size: 256

- Optimizer: Adam

- Adam $\beta_1$: 0.1

- Adam $\beta_2$: 0.001

- Adam $\varepsilon$: $10^{-8}$

Meta-model training as described in Section 3 takes 24 hours on a single RTX-3090 (24GB).

## B.2  DATASET PREPROCESSING

Recall that our base model dataset consists of $1, 6$ million datapoints, where each datapoint is a tuple $(p, w)$ where $p$ is the tokenized rasp program (an integer vector of length $r = 128$) and $w$ is the corresponding set of transformer weights (a float vector of length $m = 65, 536$).

At dataset generation time, we filter out all base models larger than $m$ parameters (this is less than $1\%$ of all models). Before model input, we treat each set of weights as a vector of length $m$, padding to length $m$ if required (we use the pad value $0.05$). Similarly, we filter out all datapoints with a RASP program longer than $r$ when tokenized.

When Tracr compiles a RASP program, a small subset of parameters in the compiled model can be very large (>1000). For this reason, we preprocess the weights array with a symmetric log-transform that is linear close to the origin:

$$w' = \begin{cases} \text{sign}(w) \log(|w|) & \text{if } |w| > 2 \\ w \cdot (\log 2)/2 & \text{otherwise.} \end{cases}$$

This transformation is chosen to be continuous, linear in the region $[-2, 2]$, and symmetrically logarithmic elsewhere.

For input to a meta-model with an embedding dimension of width $d$, we reshape the weights vector of every example to shape $(m/d, d)$. In our case, $d = 256$. We embed the tokenized RASP program $p$ via a linear layer as is standard in language modeling. We then concatenate the weights and the RASP program, resulting in an input array of shape $(m/d + r, d)$.

# C    RASP PROGRAM DATASET

The source code of our program generator is available in the supplementary material. See Figure 6 and Figure 7 for statistics on our RASP program dataset. Dataset generation can be done entirely on CPUs. Generating the RASP dataset (including compilation) takes approximately 1,000 CPU-hours (CPU cores × hours worked), most of which is spent on Tracr-compilation.

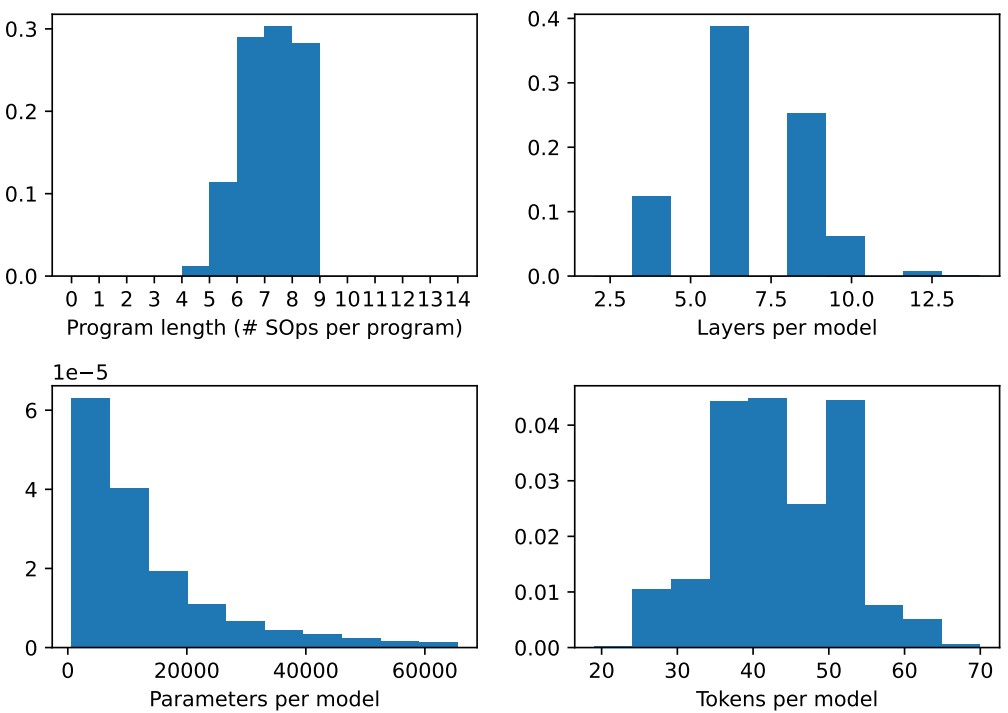

Figure 6: Statistics of the RASP program dataset used to train the decompiler.

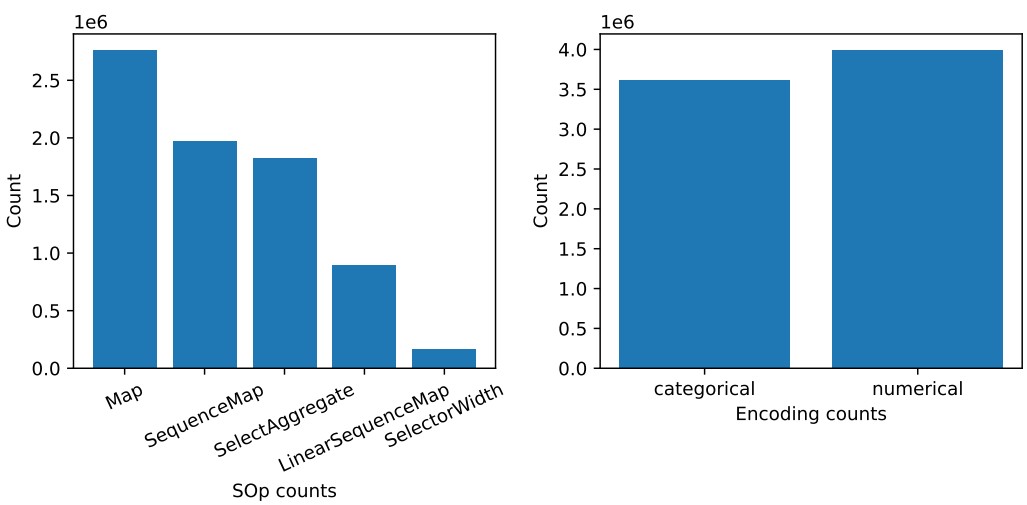

Figure 7: Statistics of the RASP program dataset used to train the decompiler.

## C.1 EXAMPLE RASP PROGRAMS

```
input: tokens, indices
all_true_sel = Select(tokens, tokens,
True)
length = SelectorWidth(all_true_sel)
return length
```

(a) Input Length

$$\text{tokens} = [\text{'h'}, \text{'e'}, \text{'l'}, \text{'l'}, \text{'o'}]$$

$$\text{all\_true\_sel} = \begin{pmatrix} T & T & T & T & T \\ T & T & T & T & T \\ T & T & T & T & T \\ T & T & T & T & T \\ T & T & T & T & T \end{pmatrix}$$

$$\text{length} = \begin{bmatrix} 5 & 5 & 5 & 5 & 5 \end{bmatrix}$$

$$\tag{1}$$

(b) Program Variables

Figure 8: Example RASP program by Lindner et al. (2023). This program uses attention operations to calculate the length of the input without the use of indices. Since the selector predicate is set to a constant the selection confusion matrix will be filled with True with equal shape to the length of tokens. When the SelectorWidth operation is applied to this the sum of each column is taken, resulting in an s-op of equal length to tokens filled with the length of the token inputs.

```
input: tokens, indices
num_l = Map(tokens, x == 'l')
prevs = Select(indices, indices, <=)
frac_prevs = Aggregate(prevs, num_l)
return frac_prevs
```

(a) Faction Previous

$$\text{tokens} = [\text{'h'}, \text{'e'}, \text{'l'}, \text{'l'}, \text{'o'}]$$

$$\text{indices} = \begin{bmatrix} 1 & 2 & 3 & 4 & 5 \end{bmatrix}$$

$$\text{num\_l} = \begin{bmatrix} 0 & 0 & 1 & 1 & 0 \end{bmatrix}$$

$$\text{prevs} = \begin{pmatrix} T & F & F & F & F \\ T & T & F & F & F \\ T & T & T & F & F \\ T & T & T & T & F \\ T & T & T & T & T \end{pmatrix}$$

$$\text{prevs} \times \text{num\_l} = \begin{pmatrix} 0 & 0 & 0 & 0 & 0 \\ 0 & 0 & 0 & 0 & 0 \\ 0 & 0 & 1 & 0 & 0 \\ 0 & 0 & 1 & 1 & 0 \\ 0 & 0 & 1 & 1 & 0 \end{pmatrix}$$

$$\text{out} = \begin{bmatrix} 0 & 0 & \frac{1}{3} & \frac{1}{2} & \frac{2}{5} \end{bmatrix}$$

$$\tag{2}$$

(b) Program Variables

Figure 9: Example RASP program by Lindner et al. (2023). This program uses a map followed by attention to calculate the fraction of tokens that were previously 'l'. The map operation simply identifies the 'l' tokens in the input. The selection matrix is independent of the tokens and is just an upper triangular matrix of shape equal to the length of the input. In the intermediate step within the aggregation operation this matrix is weighted by the s-op $num\_l$ giving a masked version of the selection matrix. Finally, the attention head averages the rows of the matrix giving the fraction of tokens seen up until that index that were 'l'.

## C.2 EXAMPLE RANDOM RASP PROGRAMS

```
input: tokens, indices
sequence_map_1 = SequenceMap(lambda x, y:  x and y, indices,
tokens)
select_1 = Select(tokens, tokens, predicate=Comparison.LEQ)
map_1 = Map(lambda x:  x != 1, tokens)
map_3 = Map(lambda x:  x + 4, sequence_map_1)
aggregate_1 = Aggregate(select_1, map_1)
map_3 = Map(lambda x:  x, aggregate_1)
sequence_map_2 = SequenceMap(lambda x, y:  x * y, map_2, map_3)
return sequence_map_2
```

Figure 10: A random RASP program sampled by our generator

```
input: tokens, indices
select_1 = Select(tokens, tokens, predicate=Comparison.GEQ)
select_2 = Select(tokens, tokens, predicate=Comparison.GT)
select_3 = Select(tokens, indices, predicate=Comparison.NEQ)
selector_width_1 = SelectorWidth(select_1)
selector_width_2 = SelectorWidth(select_2)
selector_width_3 = SelectorWidth(select_3)
sequence_map_1 = SequenceMap(lambda x, y:  x * (y + x) % 5,
selector_width_1, selector_width_2)
map_1 = Map(lambda x:  x < 0, selector_width_3)
select_4 = Select(sequence_map_1, selector_width_1,
predicate=Comparison.GEQ)
aggregate_1 = Aggregate(select_4, map_1)
return aggregate_1
```

Figure 11: A program sampled by our generator

```
input: tokens, indices
map_1 = Map(lambda x:  x + 1, indices)
select_1 = Select(map_1, indices, predicate=Comparison.GT)
selector_width_1 = SelectorWidth(select_1)
select_2 = Select(tokens, selector_width_1, predicate=Comparison.GT)
selector_width_1 = SelectorWidth(select_2)
return selector_width_1
```

Figure 12: A program sampled by our generator

```
input: tokens, indices
select_1 = Select(tokens, tokens, predicate=Comparison.GEQ)
select_2 = Select(tokens, tokens, predicate=Comparison.EQ)
selector_width_1 = SelectorWidth(select_1)
aggregate_1 = Aggregate(select_2, tokens)
select_3 = Select(tokens, aggregate_1, predicate=Comparison.EQ)
aggregate_2 = Aggregate(select_3, selector_width_1)
return aggregate_2
```

Figure 13: A program sampled by our generator

```
input: tokens, indices
sequence_map_1 = SequenceMap(lambda x, y: x * (y + 1) % 5,
indices, tokens)
sequence_map_2 = SequenceMap(lambda x, y: x or y, sequence_map_1,
indices)
return sequence_map_2
```

Figure 14: A program sampled by our generator

## D HANDCRAFTED TEST PROGRAMS

```
# sort
input: tokens, indices
smaller = Select(tokens, tokens, LT)
target = SelectorWidth(smaller)
sel = Select(target, indices, EQ)
return Aggregate(sel, tokens)
```

