# OpenReview forum: "Towards Meta-Models for Automated Interpretability"
_ICLR.cc/2025/Conference — ICLR 2025 Conference Withdrawn Submission_

### Official Review · Reviewer_W2ff · 2024-10-22

**Soundness:** 2
**Presentation:** 2
**Contribution:** 1
**Rating:** 3
**Confidence:** 3

**Summary:**

This paper trains transformers to de-compile RASP programs. It trains the meta-model (a transformer) to map transformer weights to RASP programs. It trains on randomly sampled RASP programs (1-9 operators) and evaluates the trained meta-model using i.i.d. samples. Accuracies range from 60% to 80% in various settings.

**Strengths:**

The interpretability of models is an important problem. The paper is easy to understand.

**Weaknesses:**

The most important concern is that I am not sure if training meta-models to decompile Tracr-compiled RASP programs can help interpret transformers in practice. It first assumes the functions to be learned in practice can be represented by RASP programs (at least the program shouldn't be too long to be covered in the training dataset). It also assumes the learned weights are in-distribution respective to the compilers of the RASP program so that the meta-model can generalize. It then needs to build a giant training dataset towards covering all possible RASP problems and then trains a potentially larger meta-model to learn to decompile. None of the previous assumptions are practical or intuitive to me.

Other concerns are
* The performance is not impressive. As stated by the authors, reversing Tracr is a relatively easy task at least for categorical inputs.
* The novelty is mainly limited to learning a transformer to decompile Tracr-compiled RASP programs.
* Limitations as stated by the authors: (1) Tracr-compiled weights are dissimilar to actually learned ones; (2) unlikely to cover all RASP programs in the training dataset at least using the current sampler; and so on.

**Questions:**

* How does the learned meta-model generalize to the actually learned model weights?
* Or can you train a meta-model using the actually learned model weights?

---

> ### Author Response · Authors · 2024-11-27
> **Responses to questions**
>
> Thank you for your review!
>
> > How does the learned meta-model generalize to the actually learned model weights?
> Or can you train a meta-model using the actually learned model weights?
>
> We think it is unlikely that a model trained on compiled weights could generalize directly to trained weights. To work with trained weights, the one would likely need to actually include trained base models in the training set for the meta-model. As discussed in the overall response, this is challenging because for trained models there is no easily accessible ground truth to use for training or testing.

---

> > ### Comment · Reviewer_W2ff · 2024-11-27
> >
> > Thanks the authors for the reply. Looking forward to the new experiments in future versions then.

---

### Official Review · Reviewer_LmXT · 2024-11-03

**Soundness:** 3
**Presentation:** 3
**Contribution:** 3
**Rating:** 6
**Confidence:** 4

**Summary:**

This paper explores the use of meta-models — neural networks that take other networks' parameters as input — for automated interpretability of machine learning models. The authors present a method to train transformers to map neural network weights to human-readable code, effectively creating an automated “decompiler” for neural networks. They demonstrate this approach using Tracr, a compiler that converts RASP programs (a domain-specific language for modeling transformer computations) into transformer weights.

The main contributions are:

1. Development of rasp-gen, a sampler that generates valid RASP programs, used to create a dataset of 1.6 million RASP programs and corresponding model weights
2. Training of a transformer meta-model that can recover RASP programs directly from model weights, achieving 60% accuracy for complete program reconstruction and 98.3% token-level accuracy
3. Demonstration that the trained meta-model can handle out-of-distribution examples, including successfully recovering a hand-written sorting algorithm

The authors also show their meta-model architecture outperforms previous approaches on related tasks, even when trained with less data. The work serves as a proof-of-concept for using meta-models to automate aspects of mechanistic interpretability, potentially offering a path toward more scalable neural network interpretation methods.

**Strengths:**

The paper's main strength lies in demonstrating a novel, systematic approach to automated interpretability that achieves significant results on a large dataset while laying groundwork for future developments in the field. The careful experimental design and clear presentation make the contributions accessible and (hopefully) reproducible.

Originality:

- Novel approach of using meta-models to automatically extract human-readable programs from neural network weights
- Integration of Tracr compiler with neural decompilation, effectively “reversing” the compilation process
- Method for generating large-scale training data by sampling valid RASP programs

Quality:

- Thorough empirical validation with a large dataset (1.6 million programs)
- Good quantitative results (60% accuracy on full programs, 98.3% token-level accuracy)
- Clearly presented experimental methodology
- Efficient dataset generation process (5 seconds per model on CPU)
- Additional experiments on non-sparse weights

Clarity:

- Clear problem formulation and motivation
- Well-structured presentation of methodology
- Transparent discussion of limitations and future work

Significance:

- Addresses the fundamental challenge of scalability and automated discovery in ML interpretability

**Weaknesses:**

By far the biggest weakness of this paper is the limitation to small RASP programs without much indication that the technique should be expected to generalize.  Broadly, if I can be convinced that this technique has a reasonable shot of generalizing past compiled toy examples, I would increase my score.

A substantial improvement, for example, would be to train models from scratch to match the behavior of the 1.6 million Tracr-compiled networks (separately: trained to match logits; and trained to match only the predictions / argmax output), and report numbers on decompiling these trained models to RASP programs that match their behavior.  Even though there would be no guarantee that the decompiled RASP program would implement the behavior *in the same way* as the trained network, getting a positive signal here would still be a substantial update towards direct meta-models being able to infer general behavior directly from the weights.  Even an evaluation on a couple hundred such models could be quite interesting.

More minor weaknesses:

- The characterization as of the validation on “a hand-written sorting algorithm” as “out-of-distribution with respect to the 1.6 million generated programs we use for training” (L45—47) is misleading. I would not call the sorting algorithm “out-of-distribution” just because it was removed from the training dataset.  Unless there is a (relatively natural) axis of variation (for example, length, number of variables, number of operations, number of times any given operation is used) in which the sorting algorithm can be shown to be at least 1σ away from the mean, I think it would be less misleading to say “which is not in the training distribution”.  (As an analogy, if I sample 1.6 million reals from $\mathcal{N}(0, 1)$, remove all numbers within $10^{-5}$ of 0.2, and then train a model to learn $x \mapsto x^2$, I wouldn’t say that 0.2 is “out-of-distribution” for this training.)
- Section 5 (Related Work) should include at least a brief comparison with SAEs [1] and linear probes [2], both of which can be seen as training a (very simple) model to directly interpret a neural network (albeit from the activations, rather than the weights).  [Lack of contextualization with respect to SAEs and linear probes was why I gave a "3" for presentation rather than a "4".]
- The paper would benefit from a bit more analysis of the decompilation failures.  For example, L229—230 “On a per-token level it achieves an accuracy of 98.3%” suggests that most of the failure comes from accumulation of small errors. I want to know: What is the per-token fraction of the time that the correct answer is in the top two tokens?  Top three tokens?  Top four tokens?  Top five tokens?

[1] Bricken et al., "Towards Monosemanticity: Decomposing Language Models With Dictionary Learning", Transformer Circuits Thread, 2023. https://transformer-circuits.pub/2023/monosemantic-features

[2] Guillaume Alain and Yoshua Bengio. “Understanding intermediate layers using linear classifier probes.” *arXiv*, 2016, https://arxiv.org/abs/1610.01644

**Questions:**

Questions:

- L306—307: “In addition, our meta-models tend to be larger than the base models
they are trained on by about a factor of 10-100, which would be prohibitive for very large base models.” Is there enough data to determine the scaling law here?  Is the required size linear in the base model (or the compressed base model)?  Or superlinear?
- L310 “We use a black box to interpret a black box.”  Have the authors considered applying the meta-model decompiler to itself, and seeing if the resulting RASP program is at all sensible?  This would presumably need to be combined with the program-repair scaffolding suggested below to avoid per-token errors accumulating over a length that is 10×—100× the typical program length you used, but a positive result here would again be quite interesting.

Comments:

- L229—230 “On this test dataset the decompiler is able to decompile 60% of programs without errors. On a per-token level it achieves an accuracy of 98.3%; a tokenized RASP program typically consist of between 30 and 60 tokens” Have the authors considered augmenting the model with program-repair scaffolding?  For example, given an original RASP program $P$ that is Tracr-compiled in to $C$ and decompiled into $P’$, compile $C’$ with Tracr from $P’$ and train an adversarial model to generate possible counter-examples (as suggested in L402—403 “Automated Verification of Interpretations”), train a “repair” model to take both the weights of $C$, the decompiled program $P’$, and the (input, C(input),  C’(input)) data, and suggest a new program $P’’$.
- L351—352: “in one setting fully understanding the exact algorithm implemented by a network (Nanda et al. 2023)”. Nanda et al. 2023 do not fully understand the exact algorithm implemented by the modular arithmetic models; the MLP is left mostly unexplained.  Zhong et al. 2023 [1] get closer on a simpler architecture, but even they do not explain how the MLP works.  The only works I’m aware of that can at present claim to “fully understand the exact algorithm implemented by a network” are [2] and [3].
- L400—403 “Automated Verification of Interpretations. Can a meta-model be trained to output not only a programmatic description of the base model, but also evidence or proof that this description is accurate? One approach would be to train a meta-model to adversarially suggest examples which might disprove any proposed equivalence between a model and an interpretation.” A simpler starting point would be to prove that the Tracr compilation of the output of decompilation is a close match to the original network.  If we conjecture that the activations of one network are linearly probe-able from the other network, we can train a linear probe at all of the relevant points in the network to translate activations back and forth.  Then any of the mech interp validation techniques (e.g., in order of increasing rigor: activation patching [4], causal scrubbing [5], or compact proofs [6]) could be applied to establish correspondence.  AlphaProof [7] style automation might also be possible.

Minor Comments:

- L92—93: “pred” on the LHS should be “predicate”, right?
- L243—244: “can be deterministically mapped to RASP code via
a deterministic algorithm.” using “deterministic[ally]” twice seems redundant, unless there’s something deeper going on

[1] Zhong et al. "The Clock and the Pizza: Two Stories in Mechanistic Explanation of Neural Networks." *arXiv*, 2023, https://arxiv.org/abs/2306.17844.

[2] Yip et al. “ReLU MLPs Can Compute Numerical Integration: Mechanistic Interpretation of a Non-linear Activation.” *ICML 2024 Mechanistic Interpretability Workshop*, 2024. https://openreview.net/forum?id=rngMb1wDOZ

[3] Wu et al. “Unifying and Verifying Mechanistic Interpretations: A Case Study with Group Operations.” *arXiv*, 2024, https://arxiv.org/abs/2410.07476.

[4] Stefan Heimersheim and Neel Nanda. “How to use and interpret activation patching.” *arXiv*, 2024, https://arxiv.org/abs/2404.15255.

[5] Chan et al. "Causal Scrubbing: a method for rigorously testing interpretability hypotheses." AI Alignment Forum, 2022, https://www.alignmentforum.org/posts/JvZhhzycHu2Yd57RN/causal-scrubbing-a-method-for-rigorously-testing

[6] Gross et al. “Compact Proofs of Model Performance via Mechanistic Interpretability.” *arXiv*, 2024, https://arxiv.org/abs/2406.11779.

[7] AlphaProof and AlphaGeometry teams. “AI achieves silver-medal standard solving International Mathematical Olympiad problems.” DeepMind Blog, 2024, https://deepmind.google/discover/blog/ai-solves-imo-problems-at-silver-medal-level/

---

> ### Author Response · Authors · 2024-11-27
> **Responses to review & questions**
>
> Thank you for your review.
>
> > The characterization as of the validation on “a hand-written sorting algorithm” as “out-of-distribution with respect to the 1.6 million generated programs we use for training” (L45—47) is misleading. I would not call the sorting algorithm “out-of-distribution” just because it was removed from the training dataset.
>
> The sense in which we claim it is out-of-distribution is that the generating process is different: automatic program sampling on the one hand, and writing a program by hand on the other.
>
> However, you're right that to claim it is out-of-distribution it would be good to either to confirm that 'natural' generation of the program is extremely unlikely, or use a longer hand-written program that is more clearly different from automatically sampled programs. So for now we have removed that claim from the paper.
>
> > In addition, our meta-models tend to be larger than the base models they are trained on by about a factor of 10-100, which would be prohibitive for very large base models.” Is there enough data to determine the scaling law here? Is the required size linear in the base model (or the compressed base model)? Or superlinear?
>
> This is an important question that we don't yet have an answer for. Determining scaling laws is likely the first next step after determining an appropriate benchmark - although as you mention in your review, we still have work to do in terms of improving the benchmark itself (and it is likely that the scaling laws will depend a lot on the benchmark ).
>
> > We use a black box to interpret a black box.” Have the authors considered applying the meta-model decompiler to itself, and seeing if the resulting RASP program is at all sensible?
>
> Unfortunately we expect the meta-model itself to be too OOD with respect to the training data. Not only is it larger than the base models, but also the weight statistics & distribution that result from training differ quite a bit from those that result from compilation, even when applying the compression scheme from Section 3.2. So we would strongly expect a negative result here.

---

### Official Review · Reviewer_2D3i · 2024-11-04

**Soundness:** 3
**Presentation:** 4
**Contribution:** 1
**Rating:** 3
**Confidence:** 3

**Summary:**

This paper presents an approach to automated mechanistic interpretability for Transformers by training another Transformer (the meta-model) to decode a RASP program from a given model's weights. The meta-model is trained on random RASP programs compiled to Transformers using Tracr.

The paper presents two sets of experiments. The first uses random RASP programs of up to 9 operations. The trained meta-model achieves 60% accuracy in extracting the original RASP program. The second experiment focuses on whether a meta model can extract RASP programs from non-sparse weights (since Tracr-compiled Transformers tend to have sparse weights). This is accomplished by transforming the sparse Transformer weights by 1) a random orthogonal matrix and then 2) compressing the hidden dimension via PCA. A meta-model trained on non-sparse Transformers compiled from random programs of length 5 achieves 77% accuracy.

**Strengths:**

This paper touches on a very timely problem, which is attempting to scale mechanistic interpretability by reducing the amount of manual effort required by much of the existing literature.

The work is, to the best of my knowledge, original. I am not aware of any other works that attempt to automate interpretability by training a model to decode RASP programs (or any other algorithmic representation) directly from transformer weights.

I found the writing to be generally clear. I also appreciated the limitations for being upfront and fairly comprehensive.

**Weaknesses:**

My main concerns is that the experiments are lacking in terms of demonstrating that a meta-model would be able to yield any actual insights for mechanistic interpretability. At best, the experiments have convinced me that a meta-model can invert Tracr compilation with enough data. Although I commend the authors for running the second set of experiments (with artificially dense weights), I think there is still to much of a gap between the dense weights and a "real" Transformer for the approach to have been validated.

One possibility would be to train Transformers using standard gradient descent on algorithmic outputs, then use the trained meta-model to invert them. For instance, rather than use Tracr to compile the RASP program for sorting (as done in the experiments), it would be better to *train* a Transformer to sort using data. I think validating the approach on a small number of Transformers trained (rather than compiled) to perform algorithmic tasks (on the order of 5-10) would be necessary for me to recommend acceptance.

Other concerns:
- The programs used in the experiments are all rather short, so it remains to be seen if the approach would apply to more complex / realistic domains (natural language, chess / go, or more complex algorithms)

**Questions:**

How many tokens are in the meta-model training set?

---

### Official Review · Reviewer_WQyU · 2024-11-04

**Soundness:** 4
**Presentation:** 3
**Contribution:** 2
**Rating:** 3
**Confidence:** 4

**Summary:**

In this paper, the authors train a transformer to decompile Tracr programs back into RASP format. They generate a dataset of RASP programs of length 4-8, use Tracr to compile the program into a transformer, and then train a meta-model transformer that takes the compiled transformer weights as input and autoregressively predicts the decompiled RASP program. The authors achieve 60% accuracy on a held out set, can recover a hand-written sorting program not seen during training, and get 77% decompilation accuracy on a variant of the held out set where the compiled models have a linear transformation applied to make their activations nonsparse.

**Strengths:**

The paper is well written and presented. The work is easy to understand and follow. The related work and limitations sections are good. In particular, most of the limitations I am concerned about are acknowledged in the limitations section, which is great!

**Weaknesses:**

One big weakness is the limited scope of the experiments. The authors train a transformer on a relatively small dataset of RASP programs The programs found are small, length 4–8, and the accuracy is only 60%. They only train on this dataset, and then report held out accuracy, as well as accuracy on a nonsparse held out set. I would like to see a more thorough evaluation, for example with more program sizes, or testing broader generalization abilities.

Another weakness is that I don't see any way this approach will feasibly scale to larger programs, or real world transformers. It only works because the data trained on is so small, and because we are compiling the RASP programs to generate the dataset for decompilation.

To say more, this is a fundamental limitation of this approach. Taking RASP as the domain and transformer weights as the codomain, Tracr is not anywhere close to surjective (if i understand correctly). So, any decompilation meta-model training procedure seems fundamentally unable to work on real world transformer models. This is okay if we just accept that a meta-model decompiler is only useful for Tracr-derived activations. But I don't really see the usefulness of decompiling in this case: Tracr programs are by nature created from a RASP program, so we should already know what the ground truth is.

I think the idea of using meta-models to convert a neural network into a program representation could have potential. However, training a model to do so by means of RASP + Tracr seems fundamentally limited.

Even if I accept this as a research direction, I think the present work could be more thorough in its experiments and insight. As currently presented, there is really only one dataset (the generated one) and two results (the held out set performance and the non-sparse held out set performance). I think there is a higher bar for ICLR than this amount of inquiry into a research area.

**Questions:**

The most interesting finding of this paper to me is that the meta-model recovers 77% of program on the non-sparse activations test set. It seems like such a strong train/test generalization split. Is there any intuition for why the transformer can generalize in this case? Does this hold in general cases — evaluating on a linear transformation of the input data yielding the same result? It seems too good to be true.

---

> ### Author Response · Authors · 2024-11-27
> **Response to question**
>
> Thanks for your review.
>
> Responding to your question:
> > The most interesting finding of this paper to me is that the meta-model recovers 77% of program on the non-sparse activations test set. It seems like such a strong train/test generalization split. Is there any intuition for why the transformer can generalize in this case?
>
> To clarify, in Section 3.2 we train on non-sparse weights and test on an i.i.d. test set. You're right that otherwise it would be extremely surprising if the model generalized. This wasn't clear from our description - we have uploaded a revision where this is mentioned explicitly.

---

> > ### Comment · Reviewer_WQyU · 2024-11-27
> >
> > I see, thank you for the clarification!

---

### Author Response · Authors · 2024-11-27
**Overall response**

Thank you for your excellent and thorough reviews.

All reviewers agree that the core limitation of our paper is that while our approach works on models compiled via Tracr (including on those subsequently compressed into models with non-sparse weights), we do not provide sufficient evidence that the same approach will also work to recover algorithms learned by models trained in a realistic setting.

We agree with the reviewers, and indeed this limitation is the motivation for re-running our experiments on models compressed to avoid sparsity and disentangled representations (Section 3.2). However, this experiment still involves base models that are compiled, not trained, and so it falls short of resolving the limitation.

We thank reviewers 2D3i and LmXT for suggesting follow-up experiments to gather evidence on whether our method works on trained models.

2D3i says:
> One possibility would be to train Transformers using standard gradient descent on algorithmic outputs, then use the trained meta-model to invert them. For instance, rather than use Tracr to compile the RASP program for sorting (as done in the experiments), it would be better to train a Transformer to sort using data. I think validating the approach on a small number of Transformers trained (rather than compiled) to perform algorithmic tasks (on the order of 5-10) would be necessary for me to recommend acceptance.

LmXT:
> A substantial improvement, for example, would be to train models from scratch to match the behavior of the 1.6 million Tracr-compiled networks (separately: trained to match logits; and trained to match only the predictions / argmax output), and report numbers on decompiling these trained models to RASP programs that match their behavior. Even though there would be no guarantee that the decompiled RASP program would implement the behavior in the same way as the trained network, getting a positive signal here would still be a substantial update towards direct meta-models being able to infer general behavior directly from the weights.

These are good suggestions, but there are also a number of drawbacks to this approach. We mention them here for context on why we did not run the proposed experiment originally:
- A meta-model trained on compiled (or compressed) weights is unlikely to generalize to trained weights, since the distribution of inputs is very different (e.g. weight statistics). So one would have to re-train the meta-model on a large dataset of base models obtained via training transformers to imitate RASP programs.
- There are many possible RASP programs that implement the same input-output behavior. So for any given model trained to imitate a RASP program there is no clear ground truth about which RASP program it actually implements; thus there is no clear ground truth to train or test the meta-model.
- This means that a negative result is less informative because the training objective for the meta-model is much more challenging due to the target function being ill-defined: for a given set of weights, there are many possible RASP programs that might match it.
- Conversely, a positive result is also less informative because at test time we cannot test directly against a ground-truth program; instead, we would have to check if the outputs of the trained model and the outputs of the RASP program match to >99% accuracy. This is of course a much easier task.

We now think it may be worth it to run this experiment despite the drawbacks. Of course, it would be even better to find some variant that does not suffer from the same drawbacks. Thanks again for your feedback.

---

### Note · Authors · 2024-12-12

**Comment:**

Given the concern shared by all reviewers that our experiments don't provide enough evidence that meta-models are helpful when applied to trained (rather than compiled) models, we withdraw this paper. Thanks to all the reviewers for their thorough feedback!

**Withdrawal Confirmation:**

I have read and agree with the venue's withdrawal policy on behalf of myself and my co-authors.